# High-Temperature Wear Properties of Laser Powder Directed Energy Deposited Ferritic Stainless Steel 430

**DOI:** 10.3390/mi16070752

**Published:** 2025-06-26

**Authors:** Samsub Byun, Hyun-Ki Kang, Jongyeob Lee, Namhyun Kang, Seunghun Lee

**Affiliations:** 1R&D Center, Turbo Power Tech, #107 Dasan-ro, Saha-gu, Busan 49488, Republic of Korea; ssbyun333@gmail.com (S.B.); leejy@tpt.co.kr (J.L.); 2Department of Materials Science and Engineering, Pusan National University, Busandaehak-ro 63beon-gil, Geumjeong-gu, Busan 46241, Republic of Korea; 3Welding & Joining Development Team, TESONE Co., Ltd., #37 Nakdong-daero 550beon-gil, Saha-gu, Busan 49315, Republic of Korea; sh_lee@tes-one.com

**Keywords:** additive manufacturing, directed energy deposition, ferritic stainless steel 430, wear resistance, chrome carbide precipitation, high-temperature tribology

## Abstract

Ferritic stainless steels (FSSs) have attracted considerable attention due to their excellent corrosion resistance and significantly lower cost compared with nickel-bearing austenitic stainless steels. However, the high-temperature wear behavior of additively manufactured FSS 430 has not yet been thoroughly investigated. This study aims to examine the microstructural characteristics and wear properties of laser powder directed energy deposition (LP-DED) FSS 430 fabricated under varying laser powers and hatch distances. Wear testing was conducted at 25 °C and 300 °C after subjecting the samples to solution heat treating at 815 °C and 980 °C for 1 h, followed by forced fan cooling. For comparison, an AISI 430 commercial plate was also tested under the same test conditions. The microstructural evolution and worn surfaces were analyzed using SEM-EDS and EBSD techniques. The wear performance was evaluated based on the friction coefficients and cross-sectional profiles of wear tracks, including wear volume, maximum depth, and scar width. The average friction coefficients (AFCs) of the samples solution heat treated at 980 °C were higher than those treated at 815 °C. Additionally, the AFCs increased with hatch distance at both testing temperatures. A strong correlation was observed between Rockwell hardness and wear resistance, indicating that higher hardness generally results in improved wear performance.

## 1. Introduction

Additive manufacturing (AM) is a digital fabrication technique that produces near-net-shape or final-shape parts with complex geometries. It employs focused energy sources such as lasers, electron beams, or plasma arcs to melt and deposit material layer by layer [1,2,3,4]. Among various AM techniques, powder bed fusion (PBF) and directed energy deposition (DED) are widely used for metal fabrication, particularly those employing laser-based heat sources [5,6,7]. Recently, laser powder directed energy deposition (LP-DED) has gained attention for its ability to fabricate complex ferritic stainless steel (FSS) 430 components with tailored microstructures and enhanced mechanical performance [8,9,10]. Compared with conventionally wrought or cast materials, AM-fabricated components often exhibit refined microstructures, directional solidification, and unique phase distributions that influence their mechanical and tribological performance. FSSs are extensively used in automotive exhaust systems [11,12,13], industrial furnaces, food industries, and household appliances [14,15] due to their excellent corrosion resistance, good thermal conductivity, and relatively low cost compared with nickel-bearing austenitic stainless steels [16,17,18].

Among them, FSS 430 is a chromium-rich, body-centered cubic (BCC) alloy known for its good oxidation resistance and thermal stability, making it suitable for moderate- to high-temperature applications. The typical mechanical properties of FSS 430 include a tensile strength of 400–500 MPa, a minimum yield strength of 270 MPa, an elongation of approximately 25%, and a Brinell hardness between 150 and 200 HBW. FSS 430 is particularly employed in various industrial applications such as steam turbine parts, pump shafts, valves, and bearings due to its advantageous combination of corrosion resistance, formability, and cost effectiveness. Notably, FSS 430 has been widely adopted for fabricating steam turbine seals, particularly labyrinth seals, owing to its excellent corrosion resistance, machinability, and mechanical stability at elevated operating temperatures [19,20].

Labyrinth seals made of FSS 430 serve as mechanical barriers that effectively minimize fluid leakage between rotating and stationary turbine components. These seals function by directing fluid through a tortuous, maze-like path, which dissipates pressure and minimizes flow. To ensure reliability under harsh service conditions, such seals must possess high corrosion resistance and sufficient mechanical integrity to prevent damage to the adjacent rotating parts, such as the rotor [21,22,23]. Although labyrinth seals are designed to operate without physical contact with the rotating components, unintentional contact can occur due to thermal expansion, rotor eccentricity, vibration, or assembly misalignment. Such contact can cause wear on both the seal teeth and the rotor surface, leading to increased steam leakage, reduced sealing efficiency, and potential rotor damage [24,25,26].

Previous studies have explored the wear properties of AISI 430 stainless steel under various conditions. For example, Eskandari et al. [27] examined the microstructure and wear behavior of friction stir processed AISI 430 under room-temperature pin-on-disk testing. They found that the superior wear resistance of the processed AISI 430 steel was attributed to grain refinement and martensite formation in the stir zone. Sampath et al. [28] conducted dry wear tests on SS 304, 409, and 430, reporting that 409M exhibited higher wear resistance than mild steel and SS 430, although it showed reduced corrosion resistance due to a lower chromium content. Dan et al. [29] studied the wear resistance of nitrided AISI 430 and found that it improved with increased nitriding time. Duraisamy et al. [30] investigated the tribological characteristics of wire arc additively manufactured austenitic stainless steel 347 under elevated temperatures and non-lubricated conditions. They found that the worn surfaces of the base material and SLM 347 SS exhibited rough plowing and plastic deformation with minimal oxide formation, indicating adhesive wear at 200 °C. At temperatures between 400 and 600 °C, the surfaces showed shallow grooves, fine oxide particles, and composite oxide layers, indicating oxidative wear. More recently, Jiang et al. [31] reviewed the effects of material composition, microstructure, manufacturing parameters, and heat treatment on wear performance.

However, to date, there have been no studies on the high-temperature wear properties of ferritic stainless steel 430 produced by the LP-DED process for the application of labyrinth seals in steam turbines, which operate at temperatures of approximately 500 °C. Therefore, this study investigates the microstructural evolution, including grain growth and carbide precipitation, and the high-temperature wear properties of LP-DED fabricated FSS 430 under various solution heat treatments.

## 2. Materials and Methods

### 2.1. Materials and Additive Manufacturing

Additive manufacturing was carried out using integrated LP-DED equipment (AM Solutions Co., Daejeon, Republic of Korea), consisting of a CNC machining center assembled with an IPG Photonics (Oxford, MA, USA) ytterbium fiber laser of 1000 W and beam size of 1.2 mm. The feedstock material was ferritic stainless steel (FSS 430) powder produced via the vacuum induction gas atomization (VIGA) process by Shanghai Truer Technology Co., Ltd. (Shanghai, China). The powder had particle sizes of 45–150 μm, a flowability of 19.65 s/g, and an apparent density of 4.06 g/cm^3^, and a tap density of 4.55 g/cm^3^ was employed, as shown in Figure 1.

The chemical compositions of the feedstock powder, the resulting LP-DED deposit, and the AISI 430 commercial plate are listed in Table 1, confirming compliance with the standard specification for AISI 430. The composition of the deposit was analyzed using a SPECTROLAB S optical emission spectrometer (San Diego, CA, USA). For the preliminary additive manufacturing trials, the geometry of the wear specimen is shown in Figure 2b, corresponding to the fabricated deposit illustrated in Figure 2a. The specimen had a rectangular shape with dimensions of 30 × 43 mm and a thickness of 5 mm, each illustrated in Figure 2c. Multiple experiments were performed to optimize the LP-DED process by varying key AM parameters, as summarized in Table 2. AISI 1045 carbon steel with dimensions of 200 × 200 × 30 mm^3^ (W × L × H) was used as the substrate to minimize thermal distortion during laser deposition. For comparison, an AISI 430 commercial plate (JFE SHOJI Co., Tokyo, Japan) was also employed to evaluate wear resistance under different solution heat treatment conditions.

LP-DED process parameters, including laser power and hatch distance, were varied to evaluate their influence on deposition and wear properties, as shown in Table 2. Figure 3a shows a custom-designed, real-time melt pool monitoring system (AM Solutions Co., Daejeon, Republic of Korea) integrated into the LP-DED process. The system consists of: (1) a melt pool shape monitoring camera used to estimate bead width and evaluate process stability based on melt pool morphology; (2) an infrared pyrometer for measuring melt pool temperature, which enables thermal accumulation analysis and provides feedback for laser power control; (3) a melt pool height monitoring module for real-time height measurement and feedback control to ensure stable 3D shape deposition.

The laser deposition strategy involved bidirectional scanning, with a 90° rotation of the laser scanning path between successive layers, in order to minimize anisotropy in microstructure and mechanical properties across the deposition planes, as illustrated in Figure 3b,c.

### 2.2. Solution Heat Treatment

After separating the AM wear samples (55 × 40 × 8 mm) from the substrates using wire cutting, the prepared LP-DED FSS 430 specimens and commercial AISI 430 plates were subjected to solution heat treatment to evaluate their microstructure, hardness, and wear properties in comparison with the as-built samples, as illustrated in Figure 4.

Two solution heat treatment conditions were employed. The first involved heating at 815 °C for 1 h at a rate of 4.5 °C/min, followed by forced fan cooling to 100 °C at a cooling rate of 30 °C/min. The second treatment was conducted at 980 °C for 1 h under the same heating rate, followed by forced fan cooling to 100 °C at a cooling rate of 27 °C/min.

### 2.3. Wear Test Procedure

Wear tests were performed using a Bruker UMT-TriboLab universal mechanical tester (Billerica, MA, USA) under dry conditions, in accordance with the ASTM G133-22 standard [32] for ball-on-flat configurations. An SUJ2 steel ball with a diameter of 7.9 mm was employed as the upper specimen, while various LP-DED and AISI commercial plate samples served as the lower specimens. Prior to testing, the contact surfaces of both the upper and lower specimens were thoroughly cleaned and inspected to ensure they were free from contamination or defects. The average surface (Ra) roughness of both the LP-DED FSS 430 and commercial plate samples was measured using a Surftest SV-2100 (Mitutoyo, Tokyo, Japan), as summarized in Table 3. The specimens were then securely mounted on the test apparatus. Separate wear tests were conducted at 25 °C and 300 °C, with each temperature stabilized for at least 5–10 min prior to test initiation. The experimental conditions included a normal load of 19.6 N, an oscillating frequency of 8.09 Hz, a stroke length of 11 mm, and a test duration of 30 min. Following each test, the specimens were re-inspected for any surface contamination or damage. Wear performance was evaluated based on the measured wear volume, maximum wear depth, wear scar width, and average coefficient of friction.

### 2.4. Characterization of as-Built, Plate FSS 430, and Solution Heat Treatment Samples

AM samples were prepared to characterize the LP-DED FSS 430 on the AISI 1045 carbon steel base plate, as shown in Figure 2. The as-built, commercial plate, and post-heat treatment samples were each sectioned into 3 pieces to enable comparative microstructural analysis along planes parallel (XY) and vertical (YZ) to the longitudinal deposition direction (X), as illustrated in Figure 5a,b.

The metallographic samples were prepared for scanning electron microscopy (SEM) by polishing with a 1 μm diamond suspension, followed by chemical etching using Vilella’s reagent (picric Acid 1 g + ethanol 100 mL + HCl 5 mL). Electrolytic polishing was subsequently conducted using a solution (methyl alcohol 90 mL+ perchloric acid 10 mL) at 28 volts for 90 s to achieve a refined surface state for electron back scatter diffraction (EBSD) analysis. Rockwell hardness tests (Wilson 2002T, Chicago, IL, USA) were carried out using a B-scale configuration, applying a 100 kgf load and an eight second dwell time on the polished planes parallel to the longitudinal deposition direction.

## 3. Results and Discussion

### 3.1. Investigation of Microstructure and Hardness

Figure 6 shows the microstructural evolutions of the LP-DED FSS 430 specimens fabricated under consistent laser process parameters, followed by various solution heat treatments. In the as-built sample (Figure 6a,d,g,j), columnar grains were observed along the build direction, with minimal precipitation, reflecting rapid solidification and limited diffusion, as reported by Byun et al. [8] and Lee et al. [33]. The solution heat treatment at 815 °C (Figure 6b,e,h,k) led to partial recrystallization, as indicated by the reduced aspect ratio of grains and the moderate precipitation of fine particles along grain boundaries and within grains. In contrast, the specimens treated at 980 °C (Figure 6c,f,i,l) exhibited fully recrystallized [12] and equiaxed grain structures, with a significant increase in both the size and density of precipitates, particularly along grain boundaries. These microstructural changes are attributed to enhanced diffusion, grain refinement, and carbide coarsening [34] at elevated temperatures, which may improve wear resistance but could potentially reduce toughness due to carbide embrittlement [35].

Figure 7 presents the SEM micrographs of the AISI 430 commercial plate specimens subjected to solution heat treatments at 815 °C (SHT-815) and 980 °C (SHT-980). In the low-magnification images parallel to the rolling direction (Figure 7a,b), the SHT-815 specimen (Figure 7a) reveals elongated grains aligned along the rolling direction, suggesting incomplete recrystallization. In contrast, the SHT-980 specimen (Figure 7b) shows more equiaxed and randomly oriented grains, indicating extensive recrystallization and grain structure homogenization at the higher temperature. The high-magnification SEM images (Figure 7c,d) further illustrate the grain boundary characteristics. The SHT-815 specimen (Figure 7c) exhibits relatively clean grain interiors with fine precipitates along the grain boundaries, indicative of limited diffusion during the lower-temperature treatment. The SHT-980 specimen (Figure 7d) displays broader and more defined grain boundaries, along with signs of intragranular features, likely reflecting enhanced boundary mobility and precipitate coarsening due to increased thermal energy. The vertical-to-rolling orientation (Figure 7e,f) shows a similar trend. The SHT-815 specimen (Figure 7e) retains its deformed grain structure, while the SHT-980 specimen (Figure 7f) demonstrates fully recrystallized, equiaxed grains, supporting the observations from the parallel orientation. At higher magnification (Figure 7j,k), SHT-815 again shows partial boundary definition and finer grain structures (Figure 7j), while SHT-980 (Figure 7i) exhibits clearer grain boundaries with well-separated grains and visible second-phase particles or precipitates. These results confirm that solution heat treatment at 980 °C promotes complete recrystallization and improved grain boundary development in AISI 430, while the 815 °C treatment leads to only partial recrystallization, retaining deformation-induced features. The morphological transition, observed consistently in both orientations, validates the effectiveness of higher-temperature treatment for enhancing microstructural stability and uniformity in ferritic stainless steel.

Figure 8 presents the EBSD analyses of the AISI 430 commercial plate specimens subjected to solution heat treatments at 815 °C (SHT-815) and 980 °C (SHT-980). The IQ maps (Figure 8a,b) reveal distinct differences in grain morphology. The SHT-815 specimen exhibits elongated grains aligned predominantly with the rolling direction, indicating partial recrystallization. In contrast, the SHT-980 specimen shows fully recrystallized and equiaxed grains with well-defined boundaries, reflecting complete recrystallization at the elevated temperature. The corresponding IPF maps (Figure 8c,d) support this observation. At 815 °C, the grains remain elongated and directionally textured, with an average grain size of 37.3 μm. After the heat treatment at 980 °C, the grains become more equiaxed and randomly oriented, with a slightly reduced average size of 33.9 μm, suggesting effective grain refinement and homogenization due to enhanced boundary mobility and recrystallization kinetics. The phase maps (Figure 8e,f) confirm that both specimens retain a single-phase BCC ferritic structure, with no evidence of martensitic transformation, demonstrating the thermal phase stability of AISI 430 under the applied conditions. The KAM maps (Figure 8g,h) provide insight into local strain distribution. The SHT-815 sample displays a higher density of local misorientations, particularly along grain boundaries, indicating the presence of residual strain and substructures from cold working. In contrast, the SHT-980 sample shows a more uniform and predominantly low KAM distribution, indicating reduced internal strain and enhanced microstructural recovery. Notably, the average KAM value for the SHT-980 sample (0.48°) was slightly higher than that of the SHT-815 sample (0.41°). This increase is attributed to the formation of new grain boundaries and the presence of martensitic grains interspersed with fine ferrite grains—highlighted by the red arrows—during the recrystallization process, as reported by Lu et al. [36]. Collectively, these EBSD results confirm that the solution heat treatment at 980 °C is more effective in promoting complete recrystallization, reducing internal strain, and refining grain morphology, while maintaining phase stability in ferritic stainless steel AISI 430.

Figure 9 presents the Rockwell hardness (HRB, 100 kgf load) of the LP-DED fabricated FSS 430 specimens and commercial AISI 430 plates under different heat treatment conditions and laser processing parameters. In Figure 9a,b, the solution heat-treated LP-DED specimens show clear trends in Rockwell hardness as a function of laser and hatch distance. In the solution heat treated at 815 °C, the hardness generally increases with increasing laser power, while wider hatch distances tend to reduce hardness due to insufficient overlap and weaker inter-track bonding. At 980 °C, the solution heat-treated specimens showed slightly higher hardness than those treated at 815 °C. This is because chromium carbide precipitation and coarsening occurred at the grain boundaries, as shown in Figure 6. Notably, at the highest laser power of 700 W, the hardness values become more consistent across all hatch distances, indicating enhanced microstructural homogenization and possible stress relief. Figure 9c compares AM specimens against commercial plate samples subjected to the same heat treatments. The SHT-980 commercial plate exhibits the highest hardness of 95 HRB, followed by the SHT-980 AM specimen treated at the same temperature. This enhancement is attributed to the cold-worked origin of the commercial plate, along with the fine-grained and uniform microstructure, low porosity, and efficient carbide precipitation during solution heat treatment. In contrast, the lower hardness of the specimen heat treated at 815 °C compared with those treated at 980 °C is attributed to the temperature being insufficient to fully promote the precipitation of strengthening phases such as chromium-rich carbides (e.g., Cr_23_C_6_).

### 3.2. Wear Performance

Figure 10 presents the SEM images of the worn surfaces of the LP-DED FSS 430 specimens fabricated under various combinations of laser power (500 W, 600 W, and 700 W) and hatch pitch (0.8 mm, 1.2 mm, and 1.4 mm). All samples were solution heat treated at 815 °C for 1 h prior to wear testing at ambient temperature (25 °C) under dry sliding conditions. It is generally recognized that oxidative wear dominates in ambient conditions, with adhesive wear also contributing. The wear track images suggest signs of oxidation and adhesion but lack clear grooves characteristic of abrasive wear. The mechanism involving oxidation film formation, growth to a critical thickness, and subsequent spallation aligns well with the observations in Figure 10 [37,38,39]. The wear surfaces are shown at both low and high magnifications to capture the general morphology and localized damage features. At 500 W, the worn surfaces show significant variation with hatch pitch. At 0.8 mm (Figure 10a,d), the surface is relatively smooth with shallow grooves, indicating mild abrasive wear. As hatch pitch increases to 1.2 mm (Figure 10b,e), more wear tracks and inter-track boundaries become apparent, suggesting a localized weakness in bonding. At 1.4 mm (Figure 10c,f), severe abrasive wear with debris accumulation and micro-plowing is observed, reflecting poor inter-track fusion and increased porosity due to insufficient energy input and wider spacing. At 600 W, the wear surfaces demonstrate improved uniformity. The specimen with a 0.8 mm hatch pitch (Figure 10g,j) shows minimal wear damage with fine, uniform grooves. At 1.2 mm (Figure 10h,k), the localized abrasive wear features are slightly more pronounced but remain moderate. At 1.4 mm (Figure 10i,l), the surface roughness increases.

At 700 W, all hatch distances (Figure 10m–r) result in smooth and uniform worn surfaces with minimal debris or plowing. The wear tracks are finer and more continuous, indicating enhanced inter-track bonding and homogeneity resulting from sufficient laser energy input. Notably, even at the 1.4 mm hatch distance (Figure 10o,r), the surface shows little evidence of severe wear, highlighting the robustness of the microstructure fabricated at this power level. These results indicate that both laser power and hatch distance significantly influence wear resistance in LP-DED FSS 430 specimens. A lower laser power combined with a wider hatch spacing leads to inferior inter-track bonding and a greater susceptibility to abrasive wear. In contrast, a higher laser power (≥600 W) ensures more consistent melting and bonding, which contributes to smoother wear surfaces and improved tribological performance. The findings confirm that optimized energy input is essential to enhance wear resistance in additive-manufactured ferritic stainless steel components.

Figure 11 presents the SEM-EDS analyses of the worn surface of LP-DED FSS 430 fabricated at a laser power of 500 W and a hatch distance of 1.4 mm, following reciprocating wear testing at 25 °C. The SEM micrograph (Figure 11a) highlights the analyzed region, which displays a rough, grooved surface morphology typical of abrasive wear. The presence of grooves and micro-plowing features suggests material removal due to hard asperity interaction under suboptimal processing conditions (low laser power and wide hatch distance). The EDS line scan across the selected area (Figure 11b) reveals spatial variations in elemental composition. Notably, the Cr and O signals exhibit localized enrichment, suggesting the formation of chromium-rich oxides within the wear track. The Fe signal fluctuates along the scan path, potentially due to surface exposure and the removal of base metal in high-friction zones. The presence of Mn and Si in lower but detectable amounts further supports the idea of tribo-oxidation processes involving multiple alloying elements. The EDS point spectrum (Figure 11c), acquired from a designated spot within the worn region, confirms the presence of Fe, Cr, O, Mn, and Si. The detection of oxygen in particular suggests oxidative wear, even at room temperature, likely induced by frictional heating at the contact interface during sliding. The spectrum supports the interpretation that the worn surface underwent tribochemical interactions, leading to oxide formation and the potential involvement of third-body wear mechanisms due to detached oxide particles or debris, as shown in Figure 10c,f. These results indicate that LP-DED FSS 430 specimens fabricated at low laser power and wide hatch spacing are more susceptible to surface oxidation and wear damage.

Figure 12 illustrates the frictional behavior of the LP-DED FSS 430 specimens fabricated under different laser powers and hatch distances, followed by solution heat treatment at 815 °C. The friction performance was evaluated under reciprocating sliding at 25 °C. In Figure 12a, the average friction coefficient (AFC) varies with both laser power and hatch pitch. At 500 W, the AFC increases as hatch distance widens, reaching the highest value at 1.4 mm. This trend suggests that wider hatch spacing at lower energy input results in poor inter-track bonding and increased surface roughness, which in turn contributes to higher friction. At 600 W, the AFCs are relatively balanced across hatch distances, indicating improved melt pool overlap and microstructural homogeneity. At 700 W, the friction coefficients are consistently lower across all hatch distances, reflecting enhanced metallurgical bonding and smoother surface characteristics due to the higher energy input during fabrication. Figure 12b shows the friction coefficient versus time curves for samples fabricated at 1.4 mm hatch distance and laser powers of ➀ 500 W, ➁ 600 W, and ➂ 700 W. The curve for 500 W exhibits greater fluctuation and a gradual upward trend, indicating that such fluctuations in friction curves during reciprocating wear tests are often associated with the spallation of tribofilms (e.g., oxide layers) [40]. At 600 W, the curve becomes more stable, with moderate friction fluctuations. At 700 W, the friction coefficient remains steady throughout the test, confirming stable contact conditions and superior surface integrity. These results demonstrate that higher laser power and optimized hatch spacing significantly reduce friction and enhance sliding stability. The findings support that adequate energy input during LP-DED processing is critical for achieving smoother surface finish, better track fusion, and improved tribological performance.

Figure 13 illustrates the cross-sectional profiles of wear tracks on the LP-DED FSS 430 specimens fabricated under varying laser powers (500 W, 600 W, and 700 W) and hatch distances (0.8 mm, 1.2 mm, and 1.4 mm). All specimens were subjected to solution heat treatment at 815 °C for 1 h prior to reciprocating wear testing at 25 °C. The wear area (yellow region), wear width, and maximum depth are quantitatively presented to evaluate the influence of the processing parameters on wear resistance. At 500 W, increasing hatch distance results in a noticeable increase in wear damage. The wear area rises from 223,004.8 μm^2^ at 0.8 mm to 260,009.5 μm^2^ at 1.4 mm, while the wear depth and width also increase. This indicates that a wider hatch distance under a low energy input causes weak inter-track bonding and a heterogeneous microstructure, making the surface more susceptible to material removal during sliding. At 600 W, a similar trend is observed, but with moderately improved wear performance. The wear area remains relatively stable across hatch distances (210,893.8–244,288.0 μm^2^), and the maximum wear depth shows slightly lower values than the 500 W counterparts. This suggests that moderate laser power provides improved melt pool formation and bonding, reducing the susceptibility to severe wear. At 700 W, the wear behavior improves significantly, particularly at hatch distances of 0.8 mm and 1.2 mm, with wear areas of 198,802.1 μm^2^ and 218,947.6 μm^2^, respectively—the lowest among all conditions. Notably, even at 1.4 mm, the wear area (260,114.9 μm^2^) remains comparable to the 500 W and 600 W cases, indicating that a higher laser energy compensates for the adverse effects of wider hatch spacing by enhancing inter-track fusion and surface integrity. These results confirm that higher laser power and optimized hatch spacing are critical to minimizing wear depth, width, and overall material loss. A low power and large hatch distance lead to poor bonding and high wear susceptibility, while a high power enhances wear resistance through improved microstructural homogeneity and densification.

Figure 14 shows the SEM images of the worn surfaces of LP-DED FSS 430 specimens fabricated under different laser powers (500 W, 600 W, and 700 W) and hatch distances (0.8, 1.2, and 1.4 mm), after solution heat treatment at 815 °C for 1 h and reciprocating wear testing at 300 °C. The figure includes both low- and high-magnification images to highlight the surface morphology and wear mechanisms. At 500 W, the worn surfaces exhibit significant degradation, especially as hatch distance increases. At 0.8 mm (Figure 14a,d), the surface shows moderate abrasive wear with some plastic deformation. At 1.2 mm (Figure 14b,e), the wear becomes more irregular, with increased surface roughness and oxide debris. At 1.4 mm (Figure 14c,f), severe surface damage and delamination-like features are visible, indicating unstable inter-track bonding and increased thermal softening due to wider spacing and insufficient energy input. At 600 W, the wear damage is visibly reduced across all hatch distances. The worn surfaces (Figure 14g–l) show smoother profiles with fewer signs of plowing or delamination. The surface at 0.8 mm (Figure 14g,j) shows the least wear, while 1.4 mm (Figure 14i,l) displays moderate roughness and particle detachment, indicating that 600 W provides a more stable microstructure with better thermal stability than 500 W. At 700 W, the worn surfaces are the smoothest among all conditions (Figure 14m–r), regardless of the hatch distance. Minimal surface disruption is observed, with fine and uniform wear tracks and no significant oxide debris or delamination. Even at a 1.4 mm hatch distance (Figure 14o,r), the wear morphology remains stable, demonstrating that a higher laser power ensures complete melting and strong inter-track bonding, which enhances high-temperature wear resistance. These observations confirm that a higher laser power, particularly 700 W, improves the thermal and mechanical stability of the LP-DED microstructure during elevated temperature wear. In contrast, a low power and wide hatch distances result in poor bonding, oxide layer formation, and surface degradation. Optimizing laser parameters is therefore critical to improving high-temperature tribological performance in ferritic stainless steel components produced by LP-DED.

Figure 15 presents the SEM-EDS analysis of the worn surface of LP-DED FSS 430 fabricated at a laser power of 500 W and a hatch distance of 1.4 mm, after reciprocating wear testing at 300 °C. The SEM image (Figure 15a) shows a worn surface characterized by irregular topography, micro-plowing, and evidence of surface oxidation or debris accumulation, indicating severe wear under elevated temperature conditions. The EDS line scan profile (Figure 15b) reveals significant elemental fluctuations across the analyzed region. Notably, chromium (Cr) and oxygen (O) exhibit localized co-enrichment, suggesting the formation of chromium-rich oxides. This indicates that tribo-oxidation is an active wear mechanism at this temperature, where frictional heating promotes the growth of surface oxides. The relatively stable presence of iron (Fe) confirms the exposure of the base matrix, while the detection of manganese (Mn) and silicon (Si) in lower concentrations suggests their incorporation into the oxide phases. The EDS point spectrum (Figure 15c) further confirms the presence of Fe, Cr, O, Mn, and Si in the worn region. The high oxygen content supports the interpretation of oxidative wear, and the detection of minor alloying elements reflects their migration or concentration in oxide layers or wear debris. The microstructural damage, combined with the oxidative features, indicates that the low laser power and wide hatch spacing used in fabrication result in reduced wear resistance and susceptibility to high-temperature degradation. These results demonstrate that inadequate energy input during LP-DED processing can lead to a heterogeneous microstructure prone to oxide formation and accelerated material removal during high-temperature sliding. It revealed that optimizing laser power and hatch distance is essential for suppressing tribo-oxidation and enhancing wear performance under thermal loading.

Figure 16 illustrates the frictional behavior of the LP-DED FSS 430 specimens fabricated under various laser powers and hatch distances after solution heat treatment at 815 °C, tested under reciprocating sliding at 300 °C. In Figure 16a, the average friction coefficients (AFCs) decrease notably with increasing laser power. At 500 W, the specimens show relatively higher AFCs, especially at the largest hatch distance (1.4 mm), indicating poor surface integrity and increased plowing due to weak inter-track bonding. As the laser power increases to 600 W and 700 W, the AFCs become lower and more consistent across all hatch distances, reflecting an improved microstructural uniformity and a smoother surface finish resulting from better melting and densification. Figure 16b presents the time-dependent friction coefficient profiles for specimens fabricated at a fixed hatch distance of 1.4 mm under three laser power conditions. The sample fabricated at ➀ 500 W exhibits a gradually increasing and fluctuating friction coefficient, indicating unstable contact and progressive surface damage. ➁ The 600 W sample shows a more stable friction profile with reduced variability. ➂ The 700 W sample maintains the lowest and most stable friction coefficient throughout the test duration, suggesting superior wear stability and reduced oxidation or third-body interaction. These results confirm that higher laser power significantly improves the high-temperature tribological performance of LP-DED FSS 430, particularly under wider hatch spacing, by enhancing surface bonding, reducing frictional fluctuations, and minimizing wear-induced instability.

Figure 17 presents the cross-sectional wear profiles of LP-DED FSS 430 specimens fabricated under varying laser powers (500 W, 600 W, and 700 W) and hatch distances (0.8, 1.2, and 1.4 mm), after solution heat treatment at 815 °C and reciprocating wear testing at 300 °C. The wear area (yellow region), track width, and maximum depth were quantified to evaluate the influence of the processing conditions on wear resistance. At 500 W, wear performance deteriorates with increasing hatch distance. The wear area increases markedly from 65,481.2 μm^2^ at 0.8 mm to 137,554.1 μm^2^ at 1.4 mm, accompanied by a greater depth and width. 

This trend reflects incomplete melting and weak inter-track bonding under insufficient laser energy, especially at wider hatch spacing. The specimens fabricated at 600 W show improved wear resistance, with more moderate increases in wear area and depth across hatch distances. The lowest wear area (75,482.9 μm^2^) and depth (58.1 μm) occur at 0.8 mm, suggesting more effective melt pool overlap and bonding at this condition. At 700 W, wear resistance is significantly enhanced. The minimum wear area (60,448.7 μm^2^) and depth (54.3 μm) are observed at a 1.4 mm hatch distance, and even at 1.2 mm, the wear area (61,302.6 μm^2^) remains lower than those of the 500 W and 600 W counterparts. These results indicate that a higher laser power promotes improved metallurgical bonding and homogeneity, thereby enhancing thermal wear resistance. These data confirm that both laser power and hatch distance have a strong influence on the wear behavior of LP-DED FSS 430 at elevated temperatures. The optimal wear resistance was achieved at 700 W with a 1.4 mm hatch distance, while inadequate energy input and excessive spacing increased the wear severity. These findings emphasize the importance of parameter optimization to ensure mechanical integrity in high-temperature applications.

Figure 18 presents the SEM images of the worn surfaces of LP-DED FSS 430 (a–d) and AISI 430 commercial plate specimens (e–h) after solution heat treatments at 815 °C and 980 °C, followed by reciprocating wear tests conducted at 25 °C and 300 °C, respectively, under identical testing conditions. At 25 °C, the LP-DED samples (Figure 18a,c) exhibit different wear behaviors depending on the heat treatment. The specimen treated at 815 °C (Figure 18a) shows rougher wear tracks with plowing and debris, indicating partial recrystallization and a less uniform microstructure. In contrast, the 980 °C-treated specimen (Figure 18c) presents smoother and more continuous wear tracks, suggesting improved surface stability due to full recrystallization and better inter-track bonding. Similarly, for the commercial plate at 25 °C (Figure 18e,g), the 980 °C-treated specimen (Figure 18g) displays fewer wear defects and a smoother morphology than the 815 °C counterpart (Figure 18e), indicating enhanced wear resistance from grain refinement and microstructural homogenization. At the elevated temperature of 300 °C, all specimens show increased surface damage due to thermal softening and oxidation. However, the benefits of high-temperature heat treatment remain evident. The LP-DED specimen treated at 980 °C (Figure 18d) shows less severe surface disruption compared with the one treated at 815 °C (Figure 18b), implying improved thermal wear stability. Likewise, the 980 °C-treated commercial plate (Figure 18h) shows more stable surface features and less adhesive wear compared with its 815 °C-treated counterpart (Figure 18f). These results demonstrate that solution heat treatment at 980 °C enhances wear resistance at both room and elevated temperatures for both AM and wrought FSS 430, by promoting full recrystallization, improved grain structure, and stronger oxidation resistance.

Figure 19 illustrates the frictional behavior of the LP-DED FSS 430 and AISI 430 commercial plate specimens after reciprocating wear tests performed at 25 °C and 300 °C, respectively. Figure 19a shows the average friction coefficients (AFCs), while Figure 19b presents the time-dependent friction profiles at 300 °C. In Figure 19a, the AFCs for all specimens tend to decrease slightly at 300 °C compared with 25 °C, likely due to the formation of lubricious oxide layers or thermal softening that reduces contact stress. Among the LP-DED samples, those treated at 980 °C exhibit lower AFCs than those treated at 815 °C, suggesting improved surface stability and microstructural uniformity after higher-temperature heat treatment. A similar trend is observed in the commercial plate specimens, where SHT-980 results in reduced friction relative to SHT-815. The friction coefficient versus time curves at 300 °C (Figure 19b) provide further insight. The LP-DED specimens (➀ and ➁) display higher initial coefficients and more fluctuation than the commercial plates (➂ and ➃), indicating greater surface instability and potential microstructural inhomogeneity from additive processing. Nevertheless, the LP-DED sample treated at 980 °C (➁) exhibits a more stable curve compared with the one treated at 815 °C (➀), again highlighting the benefits of high-temperature heat treatment. The commercial plates show smoother, more consistent friction profiles, with the SHT-980 specimen (➃) maintaining the lowest and most stable coefficient throughout the test. These results demonstrate that both elevated heat treatment temperature and base material type significantly influence friction behavior, with higher-temperature treatments improving sliding stability and reducing friction, especially under elevated temperature conditions.

Figure 20 presents the cross-sectional profiles of wear tracks on the LP-DED FSS 430 specimens and AISI 430 commercial plate specimens after reciprocating wear testing at 25 °C and 300 °C, respectively. For both the LP-DED and commercial materials, the wear areas of the specimens heat treated at both 815 °C and 980 °C were greater during wear testing at 25 °C than at 300 °C. This trend suggests that higher testing temperatures facilitate the formation of protective oxide layers, which likely contributed to reduced material loss through oxidative wear mechanisms. Notably, the AISI 430 commercial plate specimens exhibited significantly lower wear areas than the LP-DED specimens under the same heat treatment and testing conditions. This superior wear resistance was attributed to the homogeneous and refined microstructure of the rolled plate, as shown in Figure 7 and Figure 8, which retained finer grain sizes than those of the LP-DED FSS 430 specimen, as presented in Figure 6. In terms of wear depth, similar trends were observed. The maximum wear depths at 300 °C were consistently lower than those measured at 25 °C for all specimens, with the most pronounced reduction seen in the LP-DED samples heat treated at 980 °C. This indicates that solution heat treatment at 980 °C not only promoted recrystallization and carbide precipitation, but also enhanced the thermal stability of the wear surface. Overall, these results confirm the beneficial effects of higher solution heat treatment temperatures and elevated operating temperatures on high-temperature wear resistance, particularly in applications such as turbine seal components.

## 4. Conclusions

LP-DED FSS 430 specimens fabricated under various laser powers and hatch distances, along with AISI 430 commercial plate specimens, were subjected to solution heat treatments at 815 °C and 980 °C, followed by reciprocating wear testing at 25 °C and 300 °C, respectively. Based on the experimental results, the following conclusions were drawn:(a)**Microstructural refinement**: The solution heat treatment at 980 °C resulted in full recrystallization and the formation of equiaxed grains with increased chromium carbide precipitation, particularly along grain boundaries, while the heat treatment at 815 °C led to partial recrystallization with elongated grains and higher local misorientations. Both solution heat treatments preserved the ferritic BCC structure.(b)**Hardness enhancement**: A higher laser power and narrower hatch distance improved hardness uniformity; the treatment at 980 °C consistently yielded a higher hardness due to enhanced recrystallization and carbide coarsening.(c)**Wear behavior**: Wear resistance improved with increasing laser power. At 700 W, the LP-DED specimens showed smoother worn surfaces and minimal damage, indicating superior inter-track bonding. In contrast, a low laser power combined with wide hatch spacing led to poor bonding and severe wear.(d)**Frictional stability**: Friction testing at 300 °C showed that specimens fabricated with a higher laser power exhibited lower and more stable friction coefficients. The LP-DED samples produced at 700 W demonstrated the most stable sliding behavior, even at a 1.4 mm hatch distance.(e)**Effect of testing temperature**: All specimens exhibited lower wear areas and depths at 300 °C compared with 25 °C, which was attributed to the formation of protective oxide layers under elevated temperatures, which mitigated material loss through oxidative wear mechanisms.(f)**Comparison with commercial plate**: The AISI 430 commercial plate specimens consistently outperformed the LP-DED counterparts under identical heat treatment and testing conditions. This superior wear resistance was attributed to the refined and homogeneous microstructure of the rolled plate.(g)**Post processing significance**: Among the LP-DED specimens, those heat treated at 980 °C showed the most pronounced improvements in microstructural stability, friction behavior, and wear resistance. These results confirm the critical role of optimized solution heat treatment for enhancing the high-temperature performance of LP-DED ferritic stainless steel components in demanding applications, such as turbine seals or exhaust systems.

Overall, this study confirms that optimized LP-DED parameters and appropriate post-heat treatment are critical for enhancing the high-temperature wear performance of ferritic stainless steel 430. These findings offer valuable insights for the design of durable components such as turbine seals and exhaust system parts operating under thermal and tribological stress.

## Figures and Tables

**Figure 1 micromachines-16-00752-f001:**
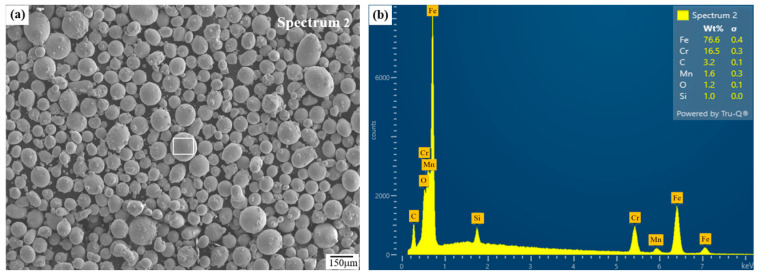
Micrographs of (**a**) feedstock FSS 430 powder and (**b**) corresponding EDS analysis [8].

**Figure 2 micromachines-16-00752-f002:**
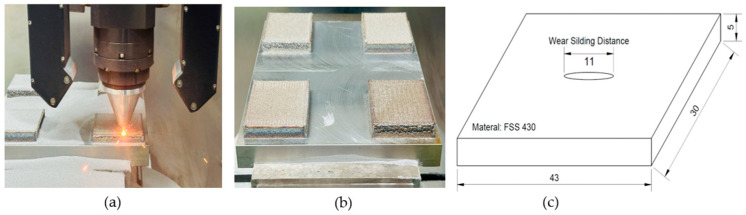
Preparations for wear testing: (**a**) laser deposition of LP-DED specimens, (**b**) fabricated wear specimens after deposition, and (**c**) schematic drawing of the wear specimen dimensions.

**Figure 3 micromachines-16-00752-f003:**
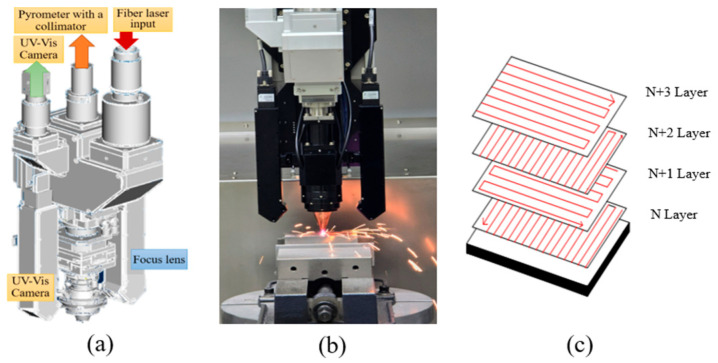
Schematics of (**a**) real-time melt pool monitoring system, (**b**) positioning of the laser optics and pyrometer system, and (**c**) scanning strategy.

**Figure 4 micromachines-16-00752-f004:**
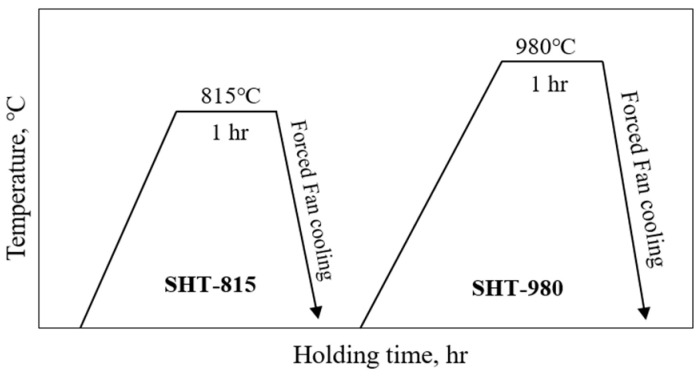
Solution heat treatment for the LP-DED FSS 430. SHT-815: solution heat treated at 815 °C for 1 h, followed by forced fan cooling. SHT-980: solution heat treated at 980 °C for 1 h, followed by forced fan cooling.

**Figure 5 micromachines-16-00752-f005:**
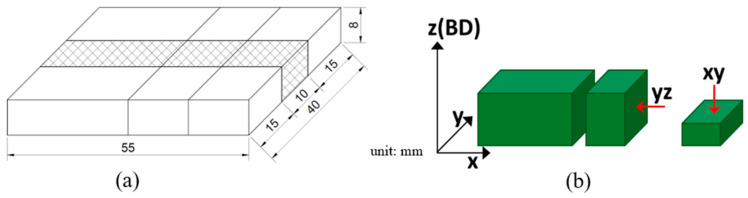
Schematic illustrations of (**a**) the deposited sample and (**b**) the area prepared for metallographic observation.

**Figure 6 micromachines-16-00752-f006:**
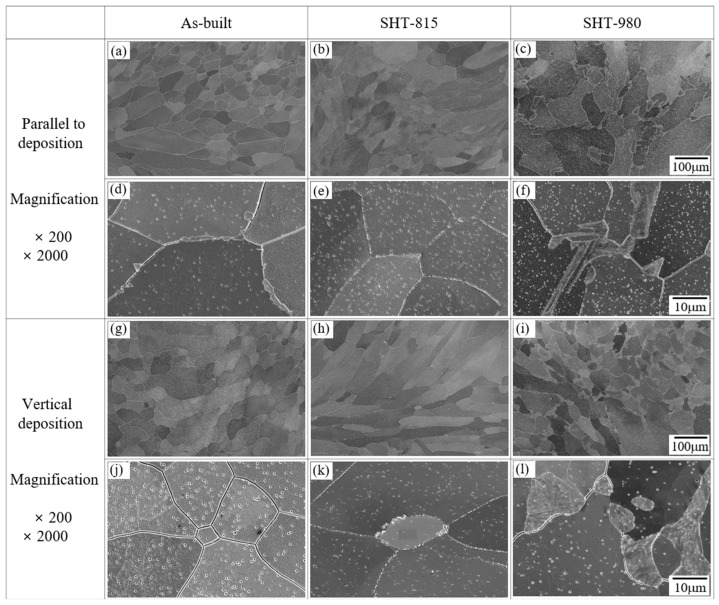
SEM images of LP-DED 430 deposits fabricated at a laser power of 600 W, scan speed of 900 mm/min, and hatch distance of 0.8 mm under various heat treatments: (**a**,**d**,**g**,**j**) as-built; (**b**,**e**,**h**,**k**) solution heat treated at 815 °C for 1 h (SHT-815); and (**c**,**f**,**i**,**l**) solution heat treated at 980 °C for 1 h (SHT-980).

**Figure 7 micromachines-16-00752-f007:**
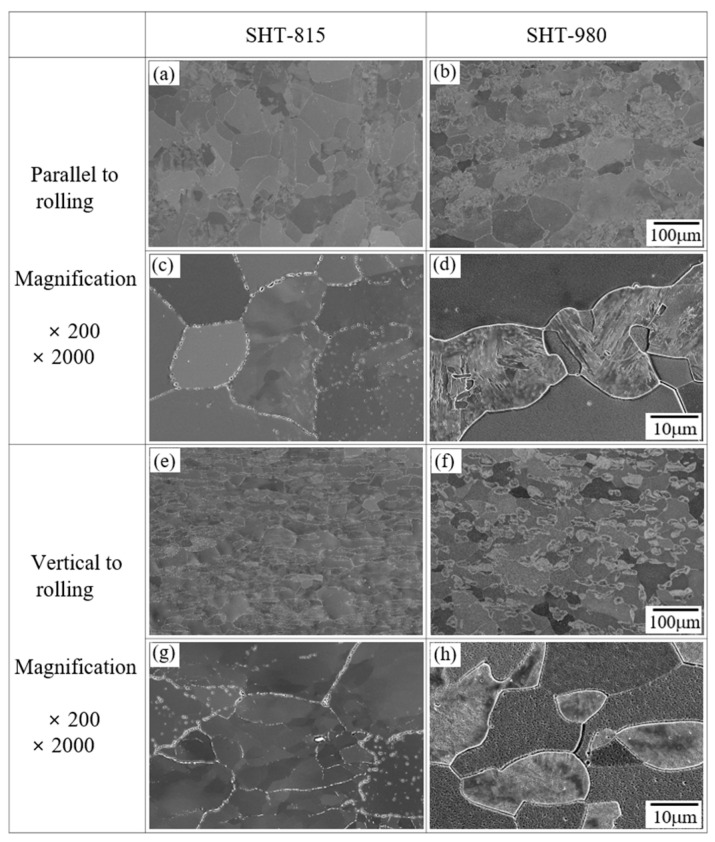
SEM images of AISI 430 commercial plate specimens: (**a**,**c**,**e**,**g**) solution heat treated at 815 °C for 1 h (SHT-815) and (**b**,**d**,**f**,**h**) solution heat treated at 980 °C for 1 h (SHT-980).

**Figure 8 micromachines-16-00752-f008:**
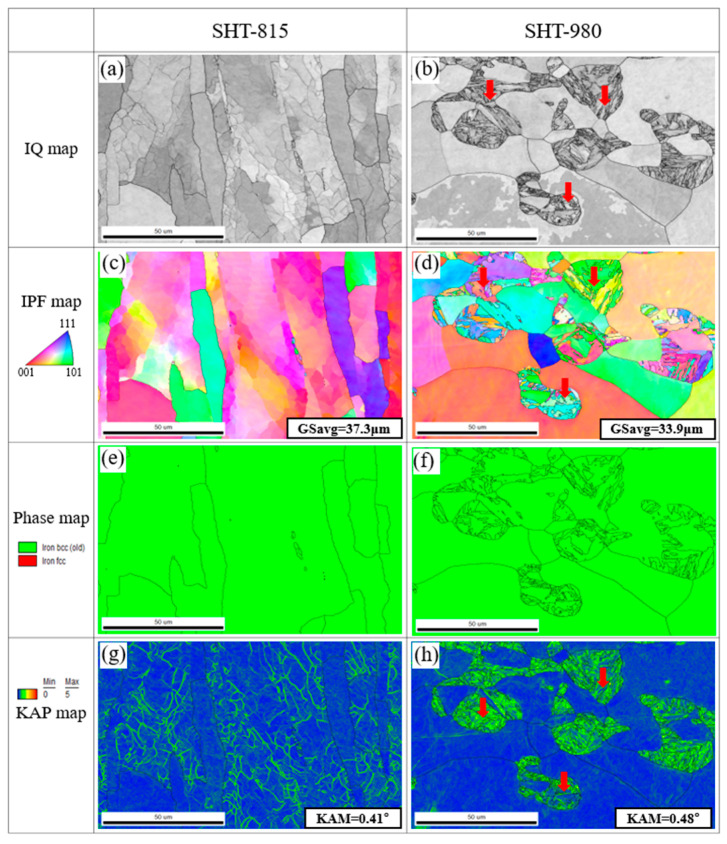
EBSD maps of AISI 430 commercial plate subjected to different heat treatments: (**a**,**c**,**e**,**g**) solution heat treated at 815 °C; (**b**,**d**,**f**,**h**) solution heat treated at 980 °C.

**Figure 9 micromachines-16-00752-f009:**
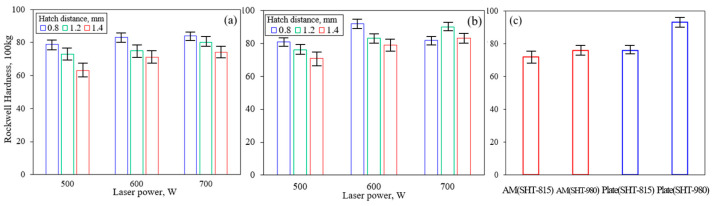
Rockwell hardness of LP-DED FSS 430 specimens and commercial plates under various process conditions: (**a**) solution heat treated at 815 °C; (**b**) solution heat treated at 980 °C; (**c**) comparison of average hardness between solution heat treated AM (SHT-815/980) and commercial plate (SHT-815/980) specimens.

**Figure 10 micromachines-16-00752-f010:**
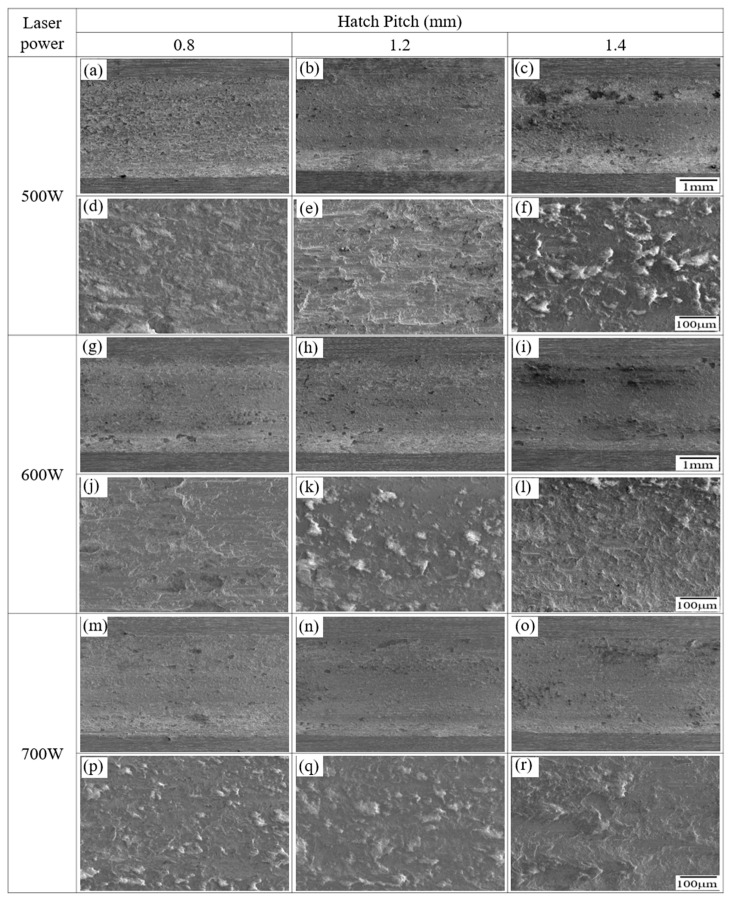
SEM images of worn surfaces of LP-DED FSS 430 specimens fabricated under various laser powers and hatch distances. Prior to wear testing, all samples were solution heat treated at 815 °C for 1 h and subsequently tested at 25 °C under dry sliding conditions (stroke length: 11 mm; normal force: 19.6 N; frequency: 8.09 Hz; duration: 30 min). (**a**,**d**) 500 W, 0.8 mm; (**b**,**e**) 500 W, 1.2 mm; (**c**,**f**) 500 W, 1.4; (**g**,**j**) 600 W, 0.8 mm; (**h**,**k**) 600 W, 0.8 mm; (**i**,**l**) 600 W, 1.4; (**m**,**p**) 700 W, 0.8 mm; (**n**,**q**) 700 W, 1.2 mm; (**o**,**r**) 700 W, 1.4 mm.

**Figure 11 micromachines-16-00752-f011:**
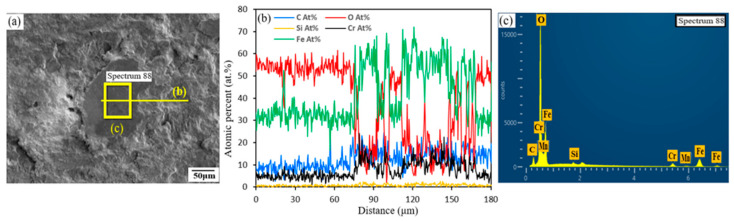
SEM-EDS analyses of the worn surface of LP-DED FSS 430 fabricated at a laser power of 500 W and a hatch distance of 1.4 mm after reciprocating wear testing at 25 °C. (**a**) SEM micrograph showing the analyzed region; (**b**) EDS line scan across the marked area; (**c**) EDS point spectrum acquired from the marked area.

**Figure 12 micromachines-16-00752-f012:**
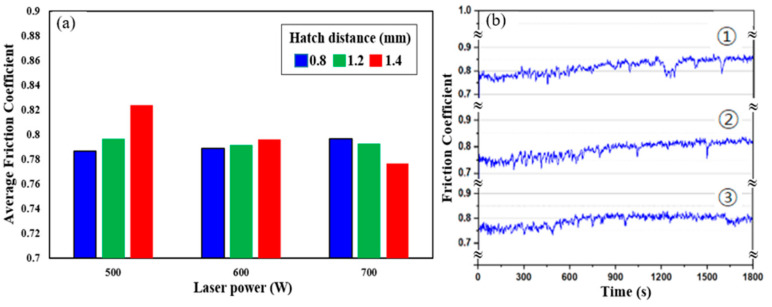
Friction behavior of LP-DED FSS 430 specimens after solution heat treatment at 815 °C. (**a**) Average friction coefficients at 25 °C under varying laser powers and hatch distances; (**b**) friction coefficient versus time at laser power of ➀ 500 W, ➁ 600 W, and ➂ 700 W with a hatch distance of 1.4 mm.

**Figure 13 micromachines-16-00752-f013:**
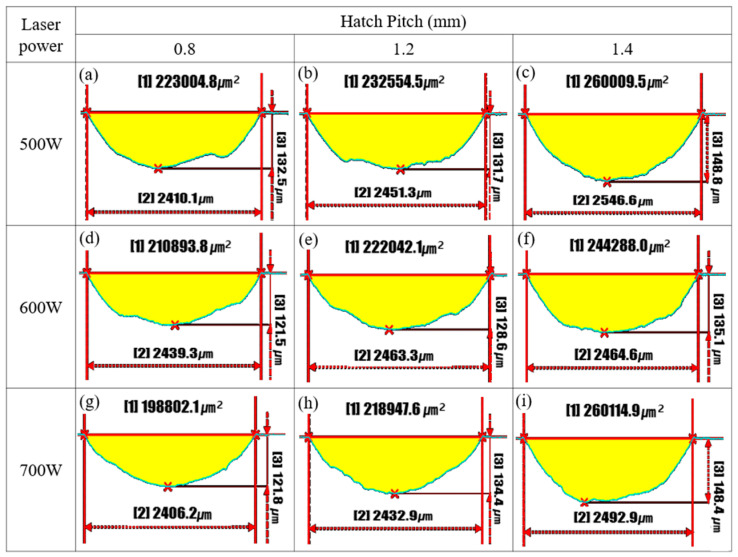
Cross-sectional profiles of wear tracks on AM FSS 430 specimens fabricated under various laser powers and hatch distances. Prior to wear testing, all samples were heat treated at 815 °C for 1 h, followed by reciprocating wear testing at 25 °C. Yellow regions indicate the worn area. [1]: wear area (μm^2^); [2]: wear track width (μm); [3]: maximum wear depth (μm). (**a**) 500 W, 0.8 mm; (**b**) 500 W, 1.2 mm; (**c**) 500W, 1.4 mm; (**d**) 600 W, 0.8 mm; (**e**) 600 W, 1.2 mm; (**f**) 600 W, 1.4 mm; (**g**) 700 W, 0.8 mm; (**h**) 700 W, 1.2 mm; (**i**) 700 W, 1.4 mm.

**Figure 14 micromachines-16-00752-f014:**
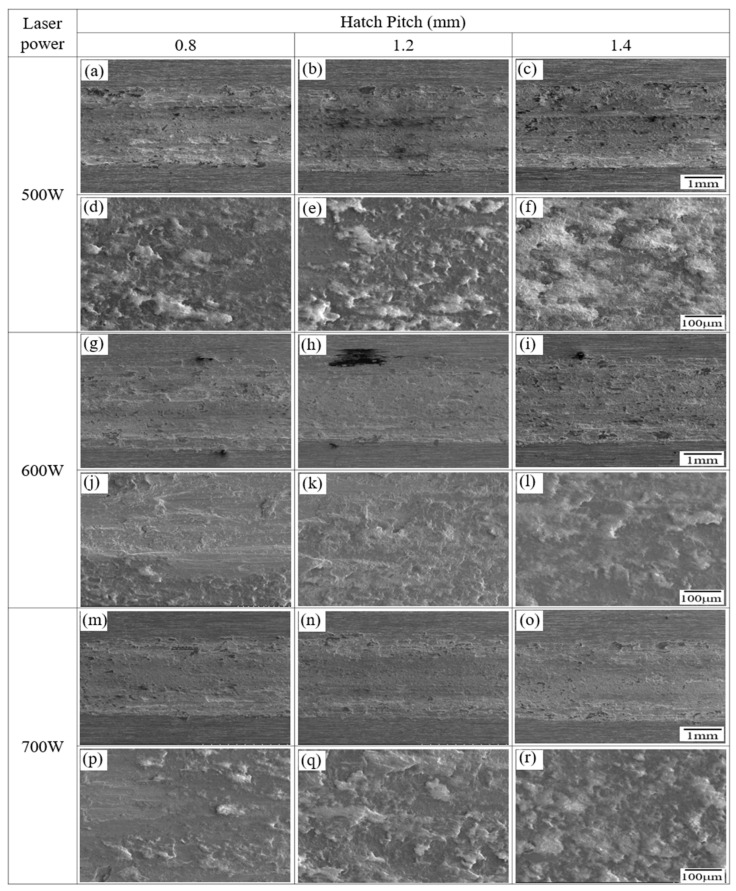
SEM images of worn surfaces of LP-DED FSS 430 specimens fabricated under various laser powers and hatch distances. Prior to wear testing, all samples were solution heat treated at 815 °C for 1 h, followed by reciprocating wear testing at 300 °C (stroke length: 11 mm; normal force: 19.6 N; frequency: 8.09 Hz; duration: 30 min). (**a,d**) 500 W, 0.8 mm; (**b,e**) 500 W, 1.2 mm; (**c,f**) 500 W, 1.4; (**g,j**) 600 W, 0.8 mm; (**h**,**k**) 600 W, 0.8 mm; (**i,l**) 600 W, 1.4; (**m,p**) 700 W, 0.8 mm; (**n,q**) 700 W, 1.2 mm; (**o,r**) 700 W, 1.4 mm.

**Figure 15 micromachines-16-00752-f015:**
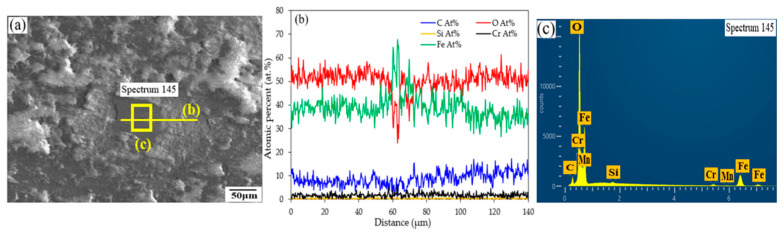
SEM-EDS analysis of the worn surface of LP-DED FSS 430 fabricated at a laser power of 500 W and a hatch distance of 1.4 mm after reciprocating wear testing at 300 °C. (**a**) SEM image showing the analyzed region; (**b**) EDS line scan profile across the marked area; (**c**) EDS point spectrum acquired from the selected spot.

**Figure 16 micromachines-16-00752-f016:**
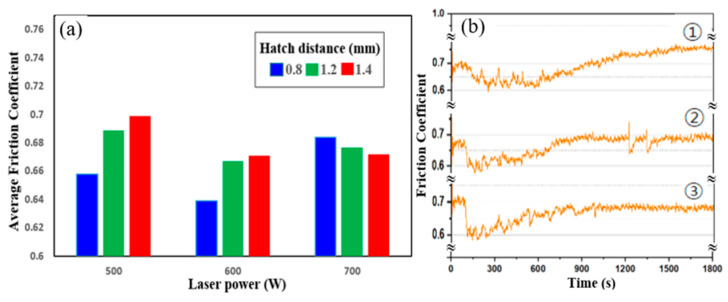
Friction behavior of LP-DED FSS 430 specimens after solution heat treatment at 815 °C. (**a**) Average friction coefficients at 300 °C under varying laser powers and hatch distances; (**b**) friction coefficient vs. time for specimens fabricated at laser power of ➀ 500 W, ➁ 600 W, and ➂ 700 W with a hatch distance of 1.4 mm.

**Figure 17 micromachines-16-00752-f017:**
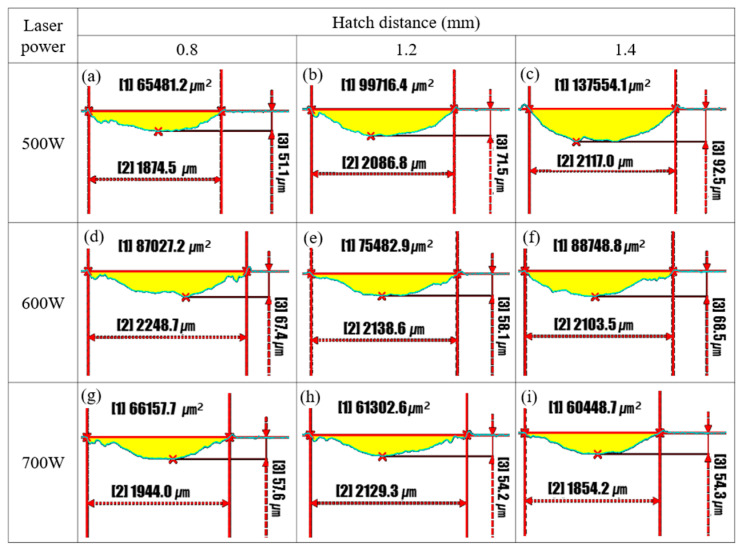
Cross-sectional profiles of wear tracks on AM FSS 430 specimens fabricated under various laser powers and hatch distances. Prior to wear testing, all samples were heat treated at 815 °C for 1 h, followed by reciprocating wear testing at 300 °C. Yellow regions indicate the worn area. [1]: wear area (μm^2^); [2]: wear track width (μm); [3]: maximum wear depth (μm). (**a**) 500 W, 0.8 mm; (**b**) 500 W, 1.2 mm; (**c**) 500W, 1.4 mm; (**d**) 600 W, 0.8 mm; (**e**) 600 W, 1.2 mm; (**f**) 600 W, 1.4 mm; (**g**) 700 W, 0.8 mm; (**h**) 700 W, 1.2 mm; (**i**) 700 W, 1.4 mm.

**Figure 18 micromachines-16-00752-f018:**
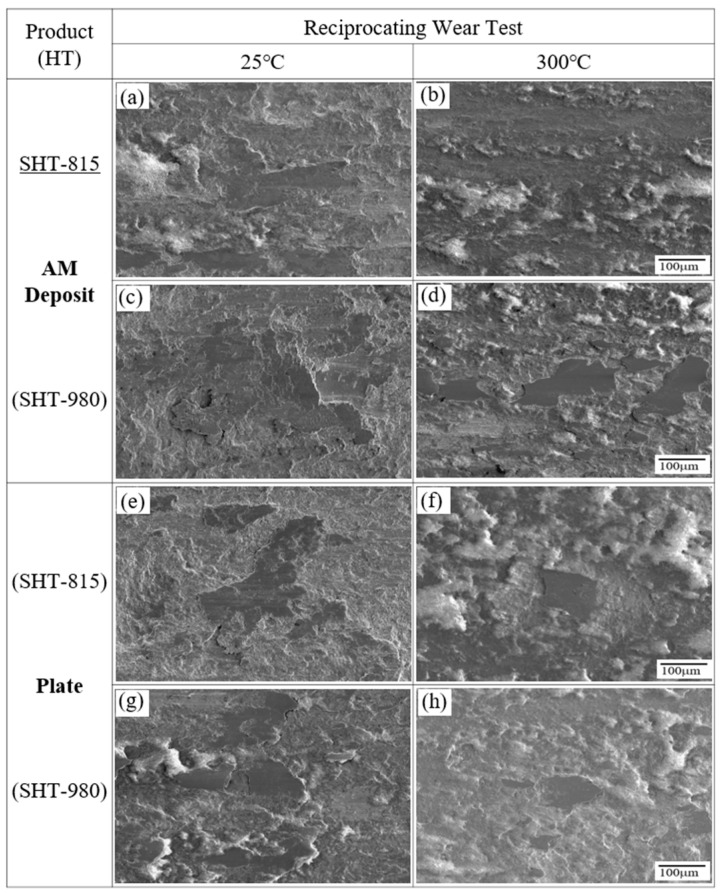
SEM images of worn surfaces of (**a**–**d**) LP-DED FSS 430 specimens and (**e**–**h**) AISI 430 commercial plate specimens. (**a**,**b**,**e**,**f**) solution heat treated at 815 °C; (**c**,**d**,**g**,**h**) solution heat treated at 980 °C for 1 h followed by reciprocating wear testing at 25 °C and 300 °C, respectively, under identical conditions (stroke length: 11 mm; normal force: 19.6 N; frequency: 8.09 Hz; duration: 30 min).

**Figure 19 micromachines-16-00752-f019:**
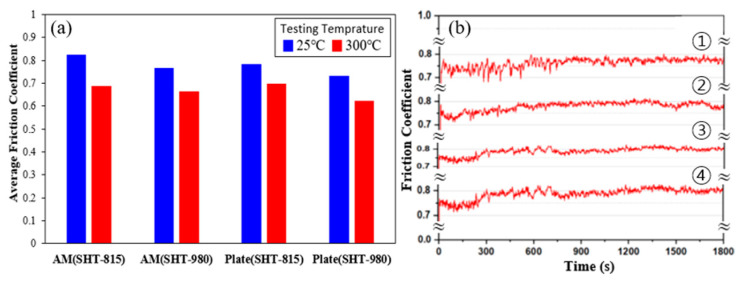
Friction behavior of LP-DED FSS 430 and AISI 430 commercial plate specimens after reciprocating wear testing at 25 °C and 300 °C, respectively: (**a**) Average friction coefficients (AFCs) under both temperatures; (**b**) friction coefficient versus time curves wear tested at 300 °C: ➀ and ➁ represent LP-DED FSS 430 specimens fabricated at a laser power of 500 W with a hatch distance of 1.4 mm; ➂ and ➃ correspond to AISI 430 commercial plate specimens.

**Figure 20 micromachines-16-00752-f020:**
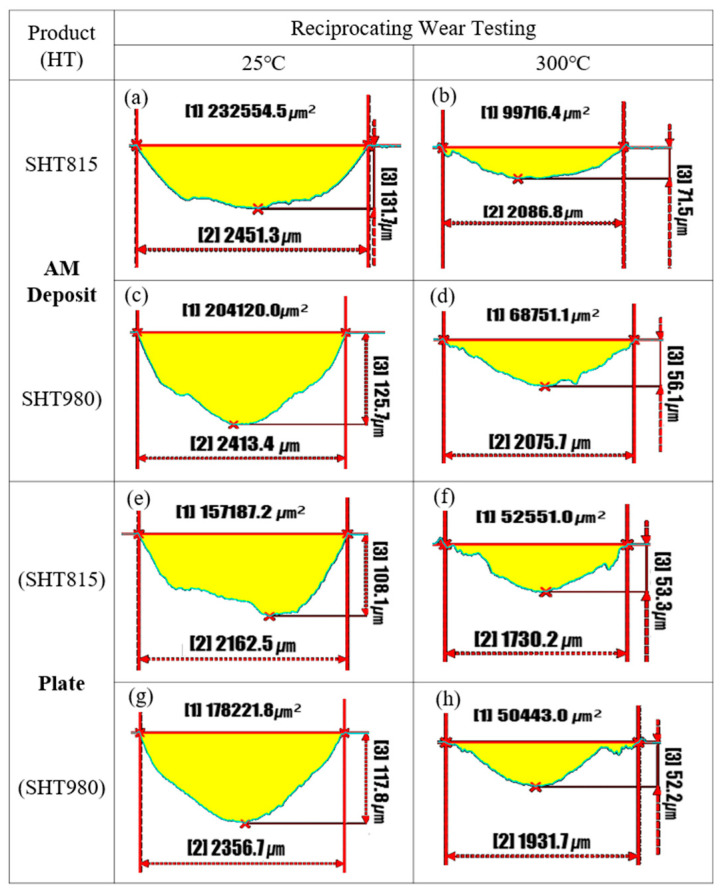
Cross-sectional profiles of wear tracks on LP-DED FSS 430 specimens and AISI 430 commercial plate specimens. Prior to wear testing, all samples were heat treated at (**a**,**b**,**e**,**f**) 815 °C and (**c**,**d**,**g**,**h**) 980 °C for 1 h, followed by reciprocating wear testing at 25 °C and 300 °C, respectively. Yellow regions indicate the worn area. [1]: wear area (μm^2^); [2]: wear track width (μm); [3]: maximum wear depth (μm).

**Table 1 micromachines-16-00752-t001:** Chemical compositions of the powder, deposit, and commercial plate studied in this work (wt.%).

	C	Si	Mn	P	S	Ni	Cr	Fe
AISI 430	<0.12	max. 1.0	max. 1.0	max. 0.04	max. 0.03	0–0.75	16–18	Bal.
Powder	max. 0.08	max. 1.0	max. 1.0	max. 0.04	max. 0.02	-	16–18	Bal.
Deposit AISI 430 plate	0.0159 0.04	0.827 0.19	0.930 0.47	0.01 0.02	0.007 0.04	0.152 0.11	17.3 16.3	Bal. Bal.

**Table 2 micromachines-16-00752-t002:** Deposition parameters of the LP-DED FSS 430.

Process	Parameter	Value
	Laser power (W)	500, 600, 700
	Scanning speed (mm/min)	900
LP-DED	Powder feed rate (g/min)	7
	Hatch distance (mm)	0.8, 1.2, 1.4
	Layer thickness (mm)	0.45

**Table 3 micromachines-16-00752-t003:** Surface roughness of LP-DED FSS 430 specimens fabricated under various laser powers and hatch distances, followed by heat treatment at 815 °C and 980 °C for 1 h.

Laser Power (W)	Hatch Distance(mm)	SHT-815	SHT-980
500	0.8	0.297	0.285
	1.2	0.338	0.310
	1.4	0.325	0.317
600	0.8	0.277	0.270
	1.2	0.311	0.301
	1.4	0.350	0.321
700	0.8	0.320	0.313
	1.2	0.317	0.305
	1.4	0.315	0.303
AISI 430 plate	-	0.325	0.311

Unit: μm.

## Data Availability

The data that support the findings of this study are available from the corresponding author upon reasonable request.

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
