# Peer review of "High-Temperature Wear Properties of Laser Powder Directed Energy Deposited Ferritic Stainless Steel 430"

_micromachines, 2025, doi:10.3390/mi16070752_

Round 1
Reviewer 1 Report
Comments and Suggestions for Authors
This study provides valuable data on the wear behavior of LP-DED FSS 430 and interprets the wear mechanisms. A major revision is recommended to clarify and substantiate the proposed wear mechanisms with more rigorous analysis.
* It must be clearly stated in the Acknowledgment section whether AI-assisted writing tools were used in manuscript preparation. Several expressions, terminologies, and structural patterns suggest the possibility of AI assistance. (This is not unethical, but it is encouraged to disclose such use for transparency.)
-
Line 163: Please clarify the cleaning method. Is cleaning sufficient to eliminate surface defects?
-
Line 165: Specify which surface roughness parameter(s) were measured (e.g., Ra, Rq, Rz, and so on)
-
Line 182: It is unclear whether the sample was sectioned into three pieces. Please revise for clarity.
-
Line 185: The term “rolling” is introduced here, but its meaning is ambiguous. Does it refer to surface rolling, powder rolling, or something else? A clear explanation of the rolling process should be added in Section 2.
-
Line 187: Use proper SI unit spacing: “1um” → “1 μm”
-
Line 189: “10mL” → “10 mL”
-
Line 192: “100kgf” → “100 kgf”
-
Figures 8 and 9 should be repositioned to match the narrative flow.
-
Figure 9: Please include standard deviation for hardness values. These values are important to support the claim regarding insufficient overlap and weak inter-track bonding. A small deviation would imply microstructural variation, whereas a large deviation would strengthen the authors’ interpretation.
-
Line 292: Similar to Comment 9, no data is provided to support the observed decrease in hardness with increased hatch distance. Please provide supporting results or relevant references.
-
Line 297: Microstructural homogenization is discussed for 700 W, but Figure 6 shows only 600 W results. Please include a representative SEM image for 700 W or cite appropriate literature.
-
Line 311: Modify phrasing to improve clarity: “under ambient temperature (25 °C) dry sliding conditions” → “at ambient temperature (25 °C) under dry sliding conditions.”
-
Figure 10: The SEM resolution is insufficient to clearly identify wear features. Replace with a higher-resolution image if available.
-
Line 316: It is generally recognized that oxidative wear dominates in ambient conditions, with adhesive wear also contributing. The wear track images suggest signs of oxidation and adhesion but lack clear grooves characteristic of abrasive wear. The mechanism involving oxidation film formation, growth to a critical thickness, and subsequent spallation aligns well with the observations in Figure 10. References: [10.1098/rspa.1980.0016, 10.1016/0043-1648(71)90004-4, 10.1080/05698196908972251].
-
Line 331: Paragraph structure is incorrect and should be revised.
-
Figures 11–17 should be repositioned to maintain logical sequencing.
-
Line 332: While inter-track bonding may contribute to wear behavior, the decreasing trend in hardness provides a more immediate and measurable explanation. The role of hardness should be considered as a primary factor influencing adhesive and oxidative wear.
-
Figure 11: Oxide spallation appears evident. The height in region (c) seems greater than in (b), possibly indicating accumulated oxide debris. A topographical analysis (e.g., 3D profilometry) would help substantiate this interpretation.
-
Line 365: Oxidative wear typically occurs via oxide film growth to a critical thickness, followed by delamination. There is limited evidence for third-body wear involving loose debris in this case.
-
Line 384: Inter-track bonding is repeatedly discussed as a critical factor, yet no measurement or quantification is provided. If considered essential, it should be experimentally evaluated or replaced with other measurable microstructural indicators.
-
Line 391: Fluctuations of friction curve in reciprocating wear tests are often linked to the spallation of tribofilms (e.g., oxide layers). The degree of oxide detachment shown in Figure 10 could be used to support this explanation.
-
Line 414: Same suggestion as Comment 18; focus on hardness as a wear-dominant factor over bonding strength alone.
-
Lines 435–461: Substantial revision is required. In high-temperature wear, dense tribofilms known as glaze layers form via frictional sintering, offering significant wear protection. Figure 14 appears to reflect such behavior. Compared to room temperature, wear tracks at high temperature show less spallation and more continuous film coverage. Suggested reference: [10.1016/0301-679X(78)90178-0].
-
Lines 468–489: This section also needs revision. High-temperature tribo-oxidation generally produces protective oxide films with sufficient mechanical strength. In the 500 W, 1.4 mm hatch sample, loose oxide debris and a lack of film consolidation suggest poor tribofilm development. While oxidation activation energy is difficult to measure directly, the authors are encouraged to cite literature to strengthen this interpretation. Suggested references: [10.1016/S0301-679X(02)00041-5, 10.1098/rspa.1963.0103, 10.1016/j.wear.2017.01.049].
- Lines 497–523: Consider briefly discussing wear trends in the context of energetic wear models or energy-based wear laws. Suggested reference: [10.1016/j.wear.2019.01.023].
Author Response
- Line 163: Please clarify the cleaning method. Is cleaning sufficient to eliminate surface defects?
- Line 165: Specify which surface roughness parameter(s) were measured (e.g., Ra, Rq, Rz, and so on)
- Revised on Line 165
3. Line 182: It is unclear whether the sample was sectioned into three pieces. Please revise for clarity.
- Revised on Line 183~187
4. Line 185: The term “rolling” is introduced here, but its meaning is ambiguous. Does it refer to surface rolling, powder rolling, or something else? A clear explanation of the rolling process should be added in Section 2.
- Line 185 has been deleted, and the relevant information can now be found in Line 116 of Section 2.
5. Line 187: Use proper SI unit spacing: “1um” → “1 μm”
- Revised
6. Line 189: “10mL” → “10 mL”
- Revised
7. Line 192: “100kgf” → “100 kgf”
- Revised
8. Figures 8 and 9 should be repositioned to match the narrative flow.
- If the reviewer is referring to the placement of subfigures labels (e.g., (a), (b), (c),...etc.), we believethe current format is acceptable, as it is consistent with the formatting used in our previous publications (Micromachines 2024, 15(7), 837; https://dog.org/10.3390/mi15070837 and Micromachines 2025, 16(5), 494; https://doi.org/10.3390/mi16050494)
9. Figure 9: Please include standard deviation for hardness values. These values are important to support the claim regarding insufficient overlap and weak inter-track bonding. A small deviation would imply microstructural variation, whereas a large deviation would strengthen the authors’ interpretation.
- Revised, as seen in Figure 9.
10. Line 292: Similar to Comment 9, no data is provided to support the observed decrease in hardness with increased hatch distance. Please provide supporting results or relevant references.
- If the reviewer is comparing Figure 9a (815 ℃) and Figure 9a (980 ℃), the improvement in the hardness after heat treatments can be understood.
11. Line 297: Microstructural homogenization is discussed for 700 W, but Figure 6 shows only 600 W results. Please include a representative SEM image for 700 W or cite appropriate literature.
- If we elaborate further on the microstructures, the manuscript will become longer and lose focus on the wear properties. Therefore, we have only stated the expected phenomena for the 700 W condition.
12. Line 311: Modify phrasing to improve clarity: “under ambient temperature (25 °C) dry sliding conditions” → “at ambient temperature (25 °C) under dry sliding conditions.”
- Revised
13. Figure 10: The SEM resolution is insufficient to clearly identify wear features. Replace with a higher-resolution image if available.
- The current image is the maximum resolution available from our SEM analysis.
14. Line 316: It is generally recognized that oxidative wear dominates in ambient conditions, with adhesive wear also contributing. The wear track images suggest signs of oxidation and adhesion but lack clear grooves characteristic of abrasive wear. The mechanism involving oxidation film formation, growth to a critical thickness, and subsequent spallation aligns well with the observations in Figure 10. References: [10.1098/rspa.1980.0016, 10.1016/0043-1648(71)90004-4, 10.1080/05698196908972251].
- We appreciate the reviewer’s valuable comments, we have incorporated the suggested paragraph, now included Line 314-319.
15. Line 331: Paragraph structure is incorrect and should be revised.
- Revised, as seen in Line 339-340
16. Figures 11–17 should be repositioned to maintain logical sequencing.
- Please refer to the response to Comment 8.
17. Line 332: While inter-track bonding may contribute to wear behavior, the decreasing trend in hardness provides a more immediate and measurable explanation. The role of hardness should be considered as a primary factor influencing adhesive and oxidative wear.
- We agree; however, our discussion here focuses on the surface characteristics.
18. Figure 11: Oxide spallation appears evident. The height in region (c) seems greater than in (b), possibly indicating accumulated oxide debris. A topographical analysis (e.g., 3D profilometry) would help substantiate this interpretation.
- Direct comparison between Figure 11(b) and (c) is not appropriate, as the observed positions are different.
19. Line 365: Oxidative wear typically occurs via oxide film growth to a critical thickness, followed by delamination. There is limited evidence for third-body wear involving loose debris in this case.
- Revised, as added Line 374
20. Line 384: Inter-track bonding is repeatedly discussed as a critical factor, yet no measurement or quantification is provided. If considered essential, it should be experimentally evaluated or replaced with other measurable microstructural indicators.
- The importance of inter-track bonding is addressed using hardness, friction coefficient, and the cross-sectional profiles of wear tracks. We believe these measurements sufficiently support our interpretation. A more detailed evaluation of inter-track bonding, including tensile test, will be presented in a forthcoming paper.
21. Line 391: Fluctuations of friction curve in reciprocating wear tests are often linked to the spallation of tribofilms (e.g., oxide layers). The degree of oxide detachment shown in Figure 10 could be used to support this explanation.
- Revised accordingly by incorporating the reviewer’s comment into Lines 397-399.
22. Line 414: Same suggestion as Comment 18; focus on hardness as a wear-dominant factor over bonding strength alone.
- We believe the current explanation appropriately addressed the influence of hardness as wear-dominant factor.
23. Lines 435–461: Substantial revision is required. In high-temperature wear, dense tribofilms known as glaze layers form via frictional sintering, offering significant wear protection. Figure 14 appears to reflect such behavior. Compared to room temperature, wear tracks at high temperature show less spallation and more continuous film coverage. Suggested reference: [10.1016/0301-679X(78)90178-0].
- We believe that the current explanation of the high-temperature is appropriate and does not require revision.
24. Lines 468–489: This section also needs revision. High-temperature tribo-oxidation generally produces protective oxide films with sufficient mechanical strength. In the 500 W, 1.4 mm hatch sample, loose oxide debris and a lack of film consolidation suggest poor tribofilm development. While oxidation activation energy is difficult to measure directly, the authors are encouraged to cite literature to strengthen this interpretation. Suggested references: [10.1016/ S0301-679X(02)00041-5, 10.1098/rspa.1963.0103, 10.1016/j.wear.2017.01.049].
- We believe that the current explanation of the high-temperature is appropriate and does not require revision.
25. Lines 497–523: Consider briefly discussing wear trends in the context of energetic wear models or energy-based wear laws. Suggested reference: [10.1016/j.wear.2019.01.023].
- We believe that the current explanation of the high-temperature is appropriate and does not require revision.
The authors thank to the reviewer for his invaluable and helpful comments, and feel certain that the manuscript has been improved by incorporating all the comments.
With best regards,
Hyun-Ki Kang
Principal Researcher
Turbo Power Tech
Reviewer 2 Report
Comments and Suggestions for Authors
Dear Authors,
Thank you for providing this interesting work related to the surface properties and the wearing resistance of Ferritic Stainless Steel 430. You may find hereafter some comments that must be addressed to improve the understanding and soundness of this work.
Abstract
During this section try to introduce why this study is important and in which industrial sector can be found useful. In general, the outputs of DED processes have not been studied extensively in different conditions such as fatigue, elevated temperature etc. However, the industrial sectors are conducting internal testing to validate the quality of the end part. For this specific material and its application area what are the requirements and why the authors are trying to extract this information?
Moreover, please try to adopt the standard terminology for laser-based DED. Fix this terminology also in the rest text, following the directions of ISO/ASTM 52926-4:2023
Introduction
Please remove the repeated “Introduction” heading at the beginning of first section.
Lines 33-41: It is suggested to present the AM processes with a more structured way, including also the main process mechanism, the type of feedstock, the production rate, the expected accuracy, costs etc.
The authors are mentioning that AM-made components can exhibit refined microstructure and case-dependent properties. Can the authors provide more insights into why this can be achieved and what characteristics of the process mechanism contribute towards this direction? Is it possible to guide the microstructure based on the selected deposition strategy? The authors may consult the following work.
- Christian Bernauer, Martina E. Sigl, Sophie Grabmann, Thomas Merk, Avelino Zapata, Michael F. Zaeh, Effects of the thermal history on the microstructural and the mechanical properties of stainless steel 316L parts produced by wire-based laser metal deposition, Materials Science and Engineering: A, Volume 889, 2024, 145862, ISSN 0921-5093, https://doi.org/10.1016/j.msea.2023.145862.
On the last paragraph of the first section try to introduce the process variables that will be tuned to achieve different part outputs and the reasoning for selecting these variables.
Materials and Methods
Why did the authors select the 90 deg infill pattern instead of 45 deg?
How did the authors decide the levels for investigated process parameters?
How did the authors use the process data generated by the monitoring devices and what kind of process control is used?
What kind of specimens are generated and how have they been selected?
Results and Discussion
Are there any rules and thresholds that can be used to characterize a microstructure as acceptable or appropriate? Did the authors follow these rules.
Conclusions
Everything is clear. However, what is the effect of process control on these results? Did the authors examined also the parts with CT scan to capture any intrinsic defects that may be linked with the reduced power?
Please include the following references for reasons of completeness:
Foteinopoulos, P., Papacharalampopoulos, A., & Stavropoulos, P. (2024). Additive manufacturing simulations: An approach based on space partitioning and dynamic 3D mesh adaptation. Additive Manufacturing Letters, 11, 100256. https://doi.org/10.1016/j.addlet.2024.100256
Thank you
Author Response
During this section try to introduce why this study is important and in which industrial sector can be found useful. In general, the outputs of DED processes have not been studied extensively in different conditions such as fatigue, elevated temperature etc. However, the industrial sectors are conducting internal testing to validate the quality of the end part. For this specific material and its application area what are the requirements and why the authors are trying to extract this information?
- For the application of labyrinth seal, high temperature tensile strength and wear resistance are critical performance requirement. We published a paper on “High-temperature properties of LP-DED additive manufactured ferritic STS 430 deposits on martensitic STS 410 base metal” (Micromachines2025, 16(5), 494; https://doi.org/10.3390/mi16050494). Building that, this manuscript focuses on hig-temperature wear properties for such industrial applications.
Moreover, please try to adopt the standard terminology for laser-based DED. Fix this terminology also in the rest text, following the directions of ISO/ASTM 52926-4:2023
- As our previous publications consistently used the term “LP-DED,” we have chosen to maintain this terminology for consistency, rather than switching to “laser based DED,” despite the ISO/ASTM 52926-4:2023 recommendation.
Introduction
Please remove the repeated “Introduction” heading at the beginning of first section.
- Revised in Line 32
Lines 33-41: It is suggested to present the AM processes with a more structured way, including also the main process mechanism, the type of feedstock, the production rate, the expected accuracy, costs etc.
The authors are mentioning that AM-made components can exhibit refined microstructure and case-dependent properties. Can the authors provide more insights into why this can be achieved and what characteristics of the process mechanism contribute towards this direction? Is it possible to guide the microstructure based on the selected deposition strategy? The authors may consult the following work.
- Christian Bernauer, Martina E. Sigl, Sophie Grabmann, Thomas Merk, Avelino Zapata, Michael F. Zaeh, Effects of the thermal history on the microstructural and the mechanical properties of stainless steel 316L parts produced by wire-based laser metal deposition, Materials Science and Engineering: A, Volume 889, 2024, 145862, ISSN 0921-5093, https://doi.org/10.1016/j.msea.2023.145862.
- We appreciate the reviewer’s suggestion. However, the referenced work focuses on wire-based laser metal deposition, whereas our study is based on powder-based laser powder-directed energy deposition (LP-DED). Due to the fundamental differences in feedstock behavior and thermal characteristics between these two processes, we believe that the cited reference is not directly applicable to our work. Therefore, we have chosen not to incorporate it at this stage.
On the last paragraph of the first section try to introduce the process variables that will be tuned to achieve different part outputs and the reasoning for selecting these variables.
Materials and Methods
Why did the authors select the 90 deg infill pattern instead of 45 deg?
- In this study, we adopted an alternating 90 deg infill pattern as part of our AM strategy. The 45 deg pattern will be considered in future work to further evaluate its effects on microstructure and wear properties.
How did the authors decide the levels for investigated process parameters?
- Among the many process parameters in LP-DED, we identified laser power and hatch distance as most critical factors influencing wear characteristics, based on our preliminary studies and literature review. Therefore, these two parameters were selected for detailed investigation in this study.
How did the authors use the process data generated by the monitoring devices and what kind of process control is used?
- We use a real-time monitoring system to observe melt pool behavior during the additive manufacturing process. However, no active process control was applied; laser power and hatch distance were maintained as predefined in the experimental plan.
What kind of specimens are generated and how have they been selected?
- The details regarding specimen types and selection criteria are provided in Section 2.1, “Materials and Additive Manufacturing” of the Materials and Methods chapter.
Results and Discussion
Are there any rules and thresholds that can be used to characterize a microstructure as acceptable or appropriate? Did the authors follow these rules.
- To our knowledge, specific rules or threshold that criteria for characterizing microstructures in this context have not yet been fully established. This study aims to explore wear performance under different conditions to help identify potential microstructural features associated with inferior performance. In particular, we highlight the importance of ensuring that the wear resistance of the labyrinth seal does not exceed that of the rotor in a steam turbine, as this could lead to damage of critical components. Our findings provide valuable data on the wear behavior of AM stainless steel at 25℃and 300℃, contributing to the development of such criteria in the future.
Conclusions
Everything is clear. However, what is the effect of process control on these results?
- Under the given process parameters, we observed that wear resistance improves with with increasing laser power and decreasing hatch distance. For more information, please refer to conclusion section.
Did the authors examined also the parts with CT scan to capture any intrinsic defects that may be linked with the reduced power?
- we did not perform CT scanning in this study, However, we appreciate the reviewer’s suggestion and will consider incorporating CT analysis in future work to investigate internal defects related to process parameters.
Please include the following references for reasons of completeness:
Foteinopoulos, P., Papacharalampopoulos, A., & Stavropoulos, P. (2024). Additive manufacturing simulations: An approach based on space partitioning and dynamic 3D mesh adaptation. Additive Manufacturing Letters, 11, 100256. https://doi.org/10.1016/j.addlet.2024.100256
The authors thank to the reviewer for his invaluable and helpful comments, and feel certain that the manuscript has been improved by incorporating all the comments.
With best regards,
Hyun-Ki Kang
Principal Researcher
Turbo Power Tech
Round 2
Reviewer 1 Report
Comments and Suggestions for Authors
The authors have addressed all reviewer comments appropriately and made the necessary revisions.
The manuscript is now clear, coherent, and scientifically sound.
I recommend it for publication in its current form.